# The All-or-Nothing Phenomenon
# in Sparse Tensor PCA

**Jonathan Niles-Weed**
Courant Insititute of Mathematical Scienes and Center for Data Science
New York University
jnw@cims.nyu.edu

**Ilias Zadik**
Center for Data Science
New York University
zadik@nyu.edu

## Abstract

We study the statistical problem of estimating a rank-one sparse tensor corrupted by additive gaussian noise, a Gaussian additive model also known as sparse tensor PCA. We show that for Bernoulli and Bernoulli-Rademacher distributed signals and *for all* sparsity levels which are sublinear in the dimension of the signal, the sparse tensor PCA model exhibits a phase transition called the *all-or-nothing phenomenon*. This is the property that for some signal-to-noise ratio (SNR) $\mathrm{SNR_c}$ and any fixed $\epsilon > 0$, if the SNR of the model is below $(1 - \epsilon)\,\mathrm{SNR_c}$, then it is impossible to achieve any arbitrarily small constant correlation with the hidden signal, while if the SNR is above $(1 + \epsilon)\,\mathrm{SNR_c}$, then it is possible to achieve almost perfect correlation with the hidden signal. The all-or-nothing phenomenon was initially established in the context of sparse linear regression, and over the last year also in the context of sparse 2-tensor (matrix) PCA, Bernoulli group testing and generalized linear models. Our results follow from a more general result showing that for any Gaussian additive model with a discrete uniform prior, the all-or-nothing phenomenon follows as a direct outcome of an appropriately defined "near-orthogonality" property of the support of the prior distribution.

## 1   Introduction

A central question in information theory and statistics is to establish the fundamental limits for recovering a planted structure in high-dimensional models. A common theme in these works is the presence of *phase transitions*, where the behavior of optimal estimators changes dramatically at critical values. An infamous example is the PCA transition, where one observes a matrix $\mathbf{Y} \in \mathbb{R}^{p \times p}$, where

$$\mathbf{Y} = \sqrt{\beta p}\,\mathbf{x}\mathbf{x}^\top + \mathbf{W}\,,$$

where $x$ is drawn uniformly from the unit sphere in $\mathbb{R}^p$ and $\mathbf{W}$ is an independent Gaussian Wigner matrix. When $\beta < 1$, then as $p \to \infty$ the leading eigenvector of $\mathbf{Y}$ is asymptotically uncorrelated with $\mathbf{x}$; on the other hand, when $\beta > 1$, the correlation between the hidden signal $\mathbf{x}$ and the leading eigenvector of $\mathbf{Y}$ remains positive as $p \to \infty$ [BBP05, FP07, BGN11].

A number of recent works establish that several sparse estimation tasks in high dimensions evince the following even more striking phase transition, called the "all-or-nothing" phenomenon [GZ17, RXZ19b, Zad19]: below a critical signal-to-noise ratio (SNR), it is impossible to achieve

any correlation with the hidden signal, but above this critical SNR, it is possible to achieve *almost perfect* correlation with the hidden signal. In other words, at any fixed SNR, one can either recover the signal perfectly, or nothing at all.

In prior work, the all-or-nothing phenomenon has been established in a smattering of different models and sparsity regimes. Understanding the extent to which this phenomenon holds more generally, beyond the models and sparsity conditions previously considered, is the main motivation of the present work. The all-or-nothing phenomenon for sparse linear regression was initially conjectured by [GZ17]. That work established that a version of this phenomenon holds for the maximum likelihood estimator (MLE): below a given threshold, the MLE does not achieve any constant correlation with the hidden signal, while above the same threshold the MLE achieves almost perfect correlation. However, this result does not rule out the existence of other estimators with better performance. Subsequently, [RXZ19b] proved that the all-or-nothing phenomenon indeed holds for sparse linear regression, when the sparsity level $k$ satisfies $k \leq p^{\frac{1}{2}-\epsilon}$ for some $\epsilon > 0$, where $p$ is the dimension of the model. Furthermore, [RXZ19a] provided generic conditions under which the phenomenon holds for sparse linear regression when $k/p = \delta > 0$ where $\delta > 0$ is a constant which shrinks to zero. Following these works, [BM19] showed that the sparse PCA model with binary and Rademacher non-zero entries also exhibits the all-or-nothing phenomenon when the sparsity level $k$ satisfies $p^{\frac{12}{13}+\epsilon} \leq k \ll p$. The "all-or-nothing" phenomenon has also recently been established in the context of the Bernoulli group testing model [TAS20]. That work proves the existence of this phenomenon in an extremely sparse setting, where $k$ scales polylogarithmically with the dimension of the model. Finally, it was recently shown in [LBM20] using analytical non-rigorous methods, that the all-or-nothing phenomenon also holds for various generalized linear models with a $k$-sparse signal, under the assumption $p^{\frac{8}{9}+\epsilon} \leq k \ll p$.

## 1.1 Contribution

In this work, in an attempt to shed some light on the fundamental reason for the existence of the all-or-nothing phenomenon, we focus on a simple Gaussian model, which we refer to as the Gaussian additive model, in which one observes a hidden signal drawn from some known prior, corrupted with additive Gaussian noise. For example, all PCA models, and in particular the sparse PCA model considered by [BM19, BMR20], are special cases of Gaussian additive models. We focus on the case where the prior is an arbitrary uniform distribution over some discrete subset of the Euclidean sphere.

We make the following contributions.

- We show that for this additive Gaussian model, the all-or-nothing phenomenon is equivalent to a simple criterion on the Kullback-Lieber divergence between the model and a null distribution with i.i.d. Gaussian entries.

- We show that, under an appropriate "near-orthogonality" condition on the prior, the all-or-nothing phenomenon always holds.

- As an application, we study *sparse tensor PCA*, in which the hidden signal is a rank-one tensor $\mathbf{x}^{\otimes d} \in (\mathbb{R}^p)^{\otimes d}$, where the entries of $x$ are $k$-sparse. We show that for both the Bernoulli and Bernoulli-Rademacher prior, all sparsity levels $k = o(p)$, and all $d \geq 2$, this model satisfies the aforementioned near-orthogonality condition, and therefore evinces the all-or-nothing phenomenon. This confirms a conjecture implicit in several prior works [BM19, BMV+18, LKZ17, PWB20, BMR20]. To the best of our knowledge this is the first result that proves the all-or-nothing phenomenon *for all sparsity levels which are sublinear in the dimension of the model.*

Omitted proofs and lemmas appear in the appendix.

## 1.2 Comparison with previous work

Our results for sparse tensor PCA are closely connected to several prior works.

**[BMV+18] and [PWB20]**   These papers study the sparse tensor PCA problem with a Bernoulli-Rademacher prior. Their focus is on optimal recovery of the hidden signal in the regime where the

sparsity satisfies $k = \gamma p$ for some constant $\gamma > 0$. [BMV$^+$18] and [PWB20] identify two thresholds, $\mathrm{SNR}_{\text{lower}}$ and $\mathrm{SNR}_{\text{upper}}$, such that below the first threshold, no constant correlation with the hidden signal is possible, while above the second threshold it is possible to obtain constant correlation with the signal. Interestingly, as $\gamma \to 0$, the two thresholds become identical. Both papers use a trick known as the conditional second moment method, and our argument in Section 4 is closely inspired by their techniques.

Our results differ from theirs in two important respects. First, though taking the sparse limit $\gamma \to 0$ is suggestive, these works do not offer a rigorous way to establish the presence of a threshold when $k = o(p)$. More importantly, the results of [BMV$^+$18] and [PWB20] elucidate the threshold between "no recovery" (zero correlation with hidden signal) and "partial recovery" (constant correlation with hidden signal) for sparse tensor PCA. By contrast, we focus on the much sharper transition between no recovery and almost perfect recovery.

**[BM19] and [BMR20]**  Unlike [BMV$^+$18] and [PWB20], [BM19] and [BMR20] study the genuinely sublinear setting when $k = o(p)$ in the special case where $d = 2$. While they prove very precise results characterizing the limiting free energy of the sparse (matrix) PCA problem, their techniques require that $k \geq p^{\frac{12}{13}+\varepsilon}$ for some $\varepsilon > 0$. Our results are less fine, insofar as we do not precisely characterize the free energy for arbitrary sparsity and SNR, but we show that the all-or-nothing phenomenon holds for a much broader range of parameters via a much simpler argument. Moreover, our results apply to the general tensor PCA problem, for all $d \geq 2$.

## 2  Main Results

### 2.1  General framework: the Gaussian Additive Model

We consider throughout the following observation model which we refer to as a *Gaussian additive model*:

$$\mathbf{Y} = \sqrt{\lambda}\mathbf{X} + \mathbf{Z}, \tag{1}$$

where $\mathbf{X} \in \mathbb{R}^N$ is drawn from a uniform discrete prior distribution $\mathrm{P}_N$ on the unit sphere in $\mathbb{R}^N$ and $\mathbf{Z} \in \mathbb{R}^N$ has i.i.d. standard Gaussian entries. We denote by $\mathrm{Q}_{\lambda,N}$ the law of $\mathbf{Y}$, where we use the subscripts $\lambda$ and $N$ to emphasize that this law depends on the signal-to-noise ratio $\lambda$ and the dimension $N$.

Given $\lambda \geq 0$, we let

$$\mathrm{MMSE}_N(\lambda) := \mathbb{E}\|\mathbf{X} - \mathbb{E}[\mathbf{X}|\mathbf{Y}]\|^2 \qquad \mathbf{Y} \sim \mathrm{Q}_{\lambda,N},$$

where $\mathbf{X}$ and $\mathbf{Y}$ are as in (1). This quantity is the smallest mean squared error achievable by any estimator of $\mathbf{X}$ based on the observation $\mathbf{Y}$. The optimal estimator $\mathbb{E}[\mathbf{X}|\mathbf{Y}]$ is commonly referred to as the Bayes-optimal estimator. The fact that $\|\mathbf{X}\| = 1$ almost surely implies that $\mathrm{MMSE}_N(\lambda) \leq 1$, since this mean-squared error is always achievable by a trivial estimator which is identically zero.

We say that a sequence of distributions $\{\mathrm{P}_N\}$ satisfies the the *all-or-nothing phenomenon* with critical SNR $\{\lambda_N\}$ if

$$\lim_{N\to\infty} \mathrm{MMSE}_N(\beta\lambda_N) = \begin{cases} 1 & \text{if } \beta < 1 \\ 0 & \text{if } \beta > 1 \,. \end{cases}$$

In other words, above some critical value, it is possible to estimate $\mathbf{X}$ nearly perfectly, but below this critical value it is not possible to estimate $\mathbf{X}$ at all, in the sense that the best estimator is no better than the trivial zero estimator.

Recall that we have assumed that $\mathrm{P}_N$ is the uniform distribution on some finite subset. Denote the cardinality of this subset by $M_N$. We assume throughout that $M_N \to \infty$ as $N \to \infty$. We also make the following assumption, which requires that the distribution $\mathrm{P}_N$ is sufficiently spread out.

**Assumption 1.**  For independent draws $\mathbf{X}, \mathbf{X}'$ from $\mathrm{P}_N$, we have

$$\lim_{t\to 1} \limsup_{N\to\infty} \frac{1}{\log M_N} \log \mathrm{P}_N^{\otimes 2}[\langle \mathbf{X}, \mathbf{X}' \rangle \geq t] \leq -1 \,.$$

In other words, Assumption 1 holds as long as the asymptotic probability that $\mathbf{X}$ and $\mathbf{X}'$ are very near each other is not much larger than the probability that $\mathbf{X} = \mathbf{X}'$.

## 2.2 Main Results: the all-or-nothing behavior for the Gaussian Additive Model

Our first main result shows that, under this assumption, there is an easy characterization of the priors which satisfy the all-or-nothing phenomenon. We denote by $D$ the Kullback-Leibler divergence (see, e.g., [PW15, Section 6]); given two probability distributions $P_1, P_2$ with $P_1$ absolutely continuous to $P_2$,

$$D(P_1 \parallel P_2) := \mathbb{E}_{P_2} \left[ \frac{dP_1}{dP_2} \log \left( \frac{dP_1}{dP_2} \right) \right].$$

**Theorem 1.** *Under Assumption 1, a sequence $\{P_N\}$ satisfies the all-or-nothing phenomenon if and only if $D(Q_{2\log M_N, N} \parallel Q_{0,N}) = o(\log M_N)$. Moreover, in this situation, we can take $\lambda_N = 2\log M_N$.*

To prove Theorem 1, we employ a well known connection between the Kullback-Liebler divergence and the MMSE, known as the I-MMSE relation (see, e.g., [GSV05]),

$$\frac{d}{d\beta} \frac{1}{\lambda_N} D(Q_{\beta\lambda_N, N} \parallel Q_{0,N}) = \frac{1}{2} - \frac{1}{2} \mathrm{MMSE}_N(\beta\lambda_N).$$

This relation implies the following characterization.

**Proposition 1.** *The all-or-nothing phenomenon holds with critical SNR $\lambda_N$ if and only if*

$$\lim_{N\to\infty} \frac{1}{\lambda_N} D(Q_{\beta\lambda_N, N} \parallel Q_{0,N}) = \frac{1}{2}(\beta - 1)_+ \qquad \forall \beta \geq 0,$$

*where $(x)_+ := \max\{x, 0\}$.*

Then, the proof of Theorem 1 exploits the fact that $\beta \mapsto \frac{1}{\lambda_N} D(Q_{\beta\lambda_N, N} \parallel Q_{0,N})$ is nonnegative, increasing, and convex. Therefore, specifying the limit for a few well-chosen values of $\beta$ is enough to establish the entire limit. We present a complete proof in Section 3.

One can naturally ask whether the above characterization yields any simple criteria for a prior to evince the all-or-nothing phenomenon. Our next result shows that the all-or-nothing phenomenon is implied by a simple condition on the *overlap* of two independent draws from $P_N$.

The condition is outlined in the following definition.

**Definition 1.** Given a non-decreasing function $r : [-1, 1] \to \mathbb{R}_{\geq 0}$, we say $\{P_N\}$ has *overlap rate function $r$* if

$$\limsup_{N\to\infty} \frac{1}{\log M_N} \log P_N^{\otimes 2}[\langle \mathbf{X}, \mathbf{X}' \rangle \geq t] \leq -r(t),$$

where $\mathbf{X}$ and $\mathbf{X}'$ are independent draws from $P_N$.

Our following result shows that a lower bound on the growth of the overlap rate function suffices to establish the all-or-nothing phenomenon.

**Theorem 2.** *Suppose that $\{P_N\}$ has overlap rate function $r$ satisfying for all $t \in [0, 1]$,*

$$r(t) \geq \frac{2t}{1+t}. \tag{2}$$

*Then $\{P_N\}$ satisfies the all-or-nothing phenomenon at $\lambda_N = 2\log M_N$.*

In words, the condition requests a particular decay condition on the upper tail of the overlap. For example, notice that the condition is trivially satisfied when the support consists of pairwise orthogonal vectors, since in that case $r(t) = 1$ for all $t \in (0, 1]$. Hence, the all-or-nothing phenomenon holds for any uniform prior distribution supported on a family of orthogonal vectors on the sphere. In the next section we present more complicated examples of prior distributions satisfying this condition. The proof of Theorem 2 is presented in Section 4.

We note that, unlike the characterization given in Theorem 1, the condition in Theorem 2 is sufficient but not necessary. To illustrate this, we consider the problem of estimating a sparse vector. Specifically, we define the *sparse vector model* to be the model given in (1), where we write $N = p$ and where $P_p$ is the uniform distribution over the subset $\{0, 1/\sqrt{k}\}^p$ with exactly $k$ nonzero entries.[1]

This problem can be viewed as a limiting ($d = 1$) case of the model presented in Section 2.3. As the following result shows, one can prove directly on the basis of Theorem 1 that this model exhibits the all-or-nothing phenomenon; however, it does *not* satisfy the overlap condition of Theorem 2.

**Proposition 2.** *For any $k = o(p)$,*

$$\limsup_{p \to \infty} \frac{1}{\log M_p} \log \mathrm{P}_p^{\otimes 2}[\langle \mathbf{X}, \mathbf{X}' \rangle \geq t] > -\frac{2t}{1+t} \quad \forall t \in (0, 1) \, ,$$

*so that $\{\mathrm{P}_p\}$ does not possess an overlap rate function satisfying (2). However, if $k = o(p)$, then the sparse vector model exhibits the all-or-nothing phenomenon with critical SNR $2k \log \left( \frac{p}{k} \right)$.*

### 2.3 Application: the sparse tensor PCA model

We apply our framework to a well-studied inference model called sparse tensor PCA, and show that it exhibits the all-or-nothing phenomenon for all sublinear sparsity levels. This model is a generalization of the sparse vector model considered in Proposition 2 to higher dimension. As we will see, when $d \geq 2$, Theorem 2 directly yields a proof of the all-or-nothing phenomenon.

For some $d \geq 2$, we define first the *tensor PCA model* to be the model given in (1), where the vectors $\mathbf{Y}$ and $\mathbf{Z}$ live in dimension $N = p^d$ and the discrete prior distribution $\mathrm{P}_N$ is supported on a subset of the vectorized $d$-tensors $\mathbf{X} = \mathbf{x}^{\otimes d}$, where this notation refers to the vector whose entry indexed by $(i_1, \ldots, i_d) \in [p]^d$ is $\mathbf{x}_{i_1} \cdots \mathbf{x}_{i_d}$. We assume the vector $\mathbf{x} \in \mathbb{R}^p$ is drawn from a discrete distribution $\widetilde{\mathrm{P}}_p$ on the unit sphere in $\mathbb{R}^p$, which induces a natural prior distribution $\mathrm{P}_N$ on the tensors $\mathbf{x}^{\otimes d}$.

We define the *sparse tensor PCA model* to be the above tensor PCA model with one of the following two prior distributions:

**Bernoulli:** $\widetilde{\mathrm{P}}_p$ is the uniform distribution over the subset of $\{0, 1/\sqrt{k}\}^p$ with exactly $k$ nonzero entries.

**Bernoulli-Rademacher:** $\widetilde{\mathrm{P}}_p$ is the uniform distribution over the subset of $\{-1/\sqrt{k}, 0, 1/\sqrt{k}\}^p$ with exactly $k$ nonzero entries.

In the supplement, we prove the following elementary bound.

**Proposition 3.** *Suppose that $k = o(p)$ and that $\widetilde{\mathrm{P}}_p$ is either Benoulli or Bernoulli-Rademacher, and let $\mathrm{P}_N$ be the induced prior distribution on $\mathbb{R}^N = \mathbb{R}^{p^d}$. Then for any $t \in [0, 1]$ it holds*

$$\lim_{N \to +\infty} \frac{1}{\log M_N} \log \mathrm{P}_N^{\otimes 2}[\langle \mathbf{X}, \mathbf{X}' \rangle \geq t] \leq -\frac{2t}{1+t} \, .$$

Combining this bound with Theorem 2 immediately yields our main result for sparse tensor PCA.

**Theorem 3.** *For any $d \geq 2$ and $k = o(p)$, the sparse tensor PCA model*

$$\mathbf{Y} = \beta \sqrt{2k \log \left( \frac{p}{k} \right)} \mathbf{x}^{\otimes d} + \mathbf{Z} \, , \quad \mathbf{x} \sim \widetilde{\mathrm{P}}_p$$

*with Bernoulli or Bernoulli-Rademacher prior exhibits the all-or-nothing phenomenon:*

$$\lim_{p \to \infty} \mathbb{E} \| \mathbf{x}^{\otimes d} - \mathbb{E}[\mathbf{x}^{\otimes d} | \mathbf{Y}] \|^2 = \begin{cases} 1 & \text{if } \beta < 1 \\ 0 & \text{if } \beta > 1 \, . \end{cases}$$

## 3 Proof of Theorem 1

In this section, we present the proof of our main equivalence, Theorem 1. As noted above, it is a consequence of Proposition 1, whose proof appears in the appendix.

We begin by giving a brief outline of the proof. It is simple to show that the function $\beta \mapsto \frac{1}{\lambda_N} \mathrm{D}(\mathrm{Q}_{\beta\lambda_N, N} \| \mathrm{Q}_{0, N})$ possesses three key properties: it is increasing, it is $\frac{1}{2}$-Lipschitz, and as $N \to +\infty$ it is bounded below by the function $\frac{1}{2}(\beta - 1)_+$. As a result, showing that it is near zero when $\beta = 1$ immediately implies that it in fact agrees with $\frac{1}{2}(\beta - 1)_+$ for *all* $\beta \geq 0$; via Proposition 1, this implies that the all-or-nothing phenomenon holds. To establish the converse claim that

the all-or-nothing phenomenon implies that the bound on $D(Q_{2\log M_N, N} \| Q_{0,N})$ holds, it suffices to show that if the all-or-nothing phenomenon holds with SNR $\{\lambda_N\}$, then we can always take $\lambda_N = 2\log M_N$. To do this, we show that the the "all" part of the all-or-nothing phenomenon implies that the the mutual information between $\mathbf{X}$ and $\mathbf{Y}$ under (1) must be approximately $\frac{\lambda_N}{2}$ when $\beta > 1$. On the other hand, if $\mathbf{X}$ can be perfectly recovered from $\mathbf{Y}$, then the mutual information must be approximately equal to the entropy of $P_N$, which is $\log M_N$. Together, these facts, combined with the geometric reasoning described above, yield the claim.

We now give the full proof. Let us first show that if $D(Q_{2\log M_N, N} \| Q_{0,N}) = o(\log M_N)$, then $\{P_N\}$ satisfies the all-or-nothing phenomenon with critical SNR equal to $2\log M_N$. Setting $\lambda_N = 2\log M_N$, we have by assumption that

$$\lim_{N\to\infty} \frac{1}{\lambda_N} D(Q_{\lambda_N, N} \| Q_{0,N}) = 0 \,, \tag{3}$$

which, since $D(Q_{\beta\lambda_N, N} \| Q_{0,N})$ is nonnegative and nondecreasing as a function of $\beta$ (Lemma 1), implies that

$$\lim_{N\to\infty} \frac{1}{\lambda_N} D(Q_{\beta\lambda_N, N} \| Q_{0,N}) = 0 \qquad \forall \beta \in [0,1] \,. \tag{4}$$

By Lemma 3, we see

$$\liminf_{N\to\infty} \frac{1}{\lambda_N} D(Q_{\beta\lambda_N, N} \| Q_{0,N}) \geq \frac{1}{2}(\beta - 1) \,.$$

However, since $\frac{1}{\lambda_N} D(Q_{\beta\lambda_N, N} \| Q_{0,N})$ is $\frac{1}{2}$-Lipschitz (Lemma 1), we have

$$\limsup_{N\to\infty} \frac{1}{\lambda_N} D(Q_{\beta\lambda_N, N} \| Q_{0,N}) \leq \frac{1}{2}|\beta - 1| + \limsup_{N\to\infty} \frac{1}{\lambda_N} D(Q_{\lambda_N, N} \| Q_{0,N}) = \frac{1}{2}|\beta - 1| \,.$$

We therefore obtain, for $\beta \geq 1$,

$$\lim_{N\to\infty} \frac{1}{\lambda_N} D(Q_{\beta\lambda_N, N} \| Q_{0,N}) = \frac{1}{2}(\beta - 1) \,.$$

Combined with (4), we obtain via Proposition 1 that $\{P_N\}$ satisfies the all-or-nothing phenomenon with critical SNR $2\log M_N$.

In the other direction, we suppose that the all-or-nothing phenomenon holds with some SNR $\lambda_N$. By Lemma 2, we can write

$$\frac{\lambda_N \beta}{2} - D(Q_{\beta\lambda_N, N} \| Q_{0,N}) = D(Q_{\beta\lambda_N, N}^{(\mathbf{X}, \mathbf{Y})} \| P_N \otimes Q_{\beta\lambda_N, N}) \,, \tag{5}$$

where $Q_{\beta\lambda_N, N}^{(\mathbf{X}, \mathbf{Y})}$ indicates the joint law of $\mathbf{X}, \mathbf{Y}$ generated according to (1).

Given an observation $\mathbf{Y}$, let us denote by $\mathbf{X}'$ a sample from the conditional distribution $P_N \mid \mathbf{Y}$. If $(\mathbf{X}, \mathbf{Y}) \sim Q_{\beta\lambda_N, N}^{(\mathbf{X}, \mathbf{Y})}$, then this induces a joint distribution on $(\mathbf{X}, \mathbf{X}')$ which we denote by $P_{\beta\lambda_N, N}$. On the other hand, if $\mathbf{X}$ and $\mathbf{Y}$ are independent, then $\mathbf{X}$ and $\mathbf{X}'$ are independent and marginally each has distribution $P_N$, so the pair $(\mathbf{X}, \mathbf{X}')$ has law $P_N^{\otimes 2}$.

Applying the data processing inequality twice, we obtain for any event $\Omega$ that

$$D(Q_{\beta\lambda_N, N}^{(\mathbf{X}, \mathbf{Y})} \| P_N \otimes Q_{\beta\lambda_N, N}) \geq D(P_{\beta\lambda_N, N} \| P_N^{\otimes 2}) \geq d(P_{\beta\lambda_N, N}(\Omega) \| P_N^{\otimes 2}(\Omega)) \,,$$

where $d$ is the binary divergence function:

$$d(\alpha_1 \| \alpha_2) := \alpha_1 \log \frac{\alpha_1}{\alpha_2} + (1 - \alpha_1) \log \frac{1 - \alpha_1}{1 - \alpha_2} \,.$$

Fix a $t \in [0, 1)$, and set
$$\Omega_t := \{(x, x') : \langle x, x' \rangle \geq t\} \,.$$

Suppose $(\mathbf{X}, \mathbf{X}') \sim P_{\beta\lambda_N, N}$. For any $\beta > 1$, the fact that the all-or-nothing phenomenon holds implies that
$$\mathbb{E}\langle \mathbf{X}, \mathbf{X}' \rangle = \mathbb{E}\langle \mathbf{X}, \mathbb{E}[\mathbf{X}|\mathbf{Y}] \rangle = 1 - o(1) \,,$$

Since $\langle \mathbf{X}, \mathbf{X}' \rangle \leq 1$ almost surely, this implies that we must also have

$$\lim_{N \to \infty} \mathrm{P}_{\beta \lambda_N, N}(\Omega_t) = 1.$$

On the other hand, Assumption 1 implies that for $t$ sufficiently close to 1,

$$\lim_{N \to \infty} \mathrm{P}_N^{\otimes 2}(\Omega_t) = 0.$$

Combining these observations, we obtain for any $\beta > 1$ and $t$ sufficiently close to 1 that

$$\limsup_{N \to \infty} \frac{1}{\lambda_N} \mathrm{D}(\mathrm{Q}_{\beta \lambda_N, N}(\mathbf{X}, \mathbf{Y}) \,\|\, \mathrm{P}_N \otimes \mathrm{Q}_{\beta \lambda_N, N}) \geq \limsup_{N \to \infty} \frac{1}{\lambda_N} d(\mathrm{P}_{\beta \lambda_N, N}(\Omega_t) \,\|\, \mathrm{P}_N^{\otimes 2}(\Omega_t))$$

$$= \limsup_{N \to \infty} \frac{1}{\lambda_N} \log \frac{1}{\mathrm{P}_N^{\otimes 2}(\Omega_t)},$$

where we justify the final limit in Lemma 4, noting that (5) implies that $\frac{1}{\lambda_N} \mathrm{D}(\mathrm{Q}_{\beta \lambda_N, N}(\mathbf{X}, \mathbf{Y}) \,\|\, \mathrm{P}_N \otimes \mathrm{Q}_{\beta \lambda_N, N})$ is bounded.

Under the all-or-nothing phenomenon, Proposition 1 and (5) imply that the left side of the above inequality is $1/2$. Combining this with Assumption 1, we obtain that for any $\delta > 0$, there exists $t \in [0, 1)$ such that for all $N$ sufficiently large,

$$\frac{1}{\lambda_N} \log \mathrm{P}_N^{\otimes 2}(\Omega_t) \geq -\frac{1}{2} - \delta$$

$$\frac{1}{\log M_N} \log \mathrm{P}_N^{\otimes 2}(\Omega_t) \leq -1 + \delta$$

In particular, we must have $\lambda_N \geq (2 - O(\delta)) \log M_N$ for all $N$ large enough, so that

$$\liminf_{N \to \infty} \frac{\lambda_N}{\log M_N} \geq 2 - O(\delta),$$

and letting $\delta \to 0$ yields

$$\liminf_{N \to \infty} \frac{\lambda_N}{\log M_N} \geq 2.$$

On the other hand, Lemma 3 implies

$$\lim_{N \to \infty} \frac{1}{\lambda_N} \mathrm{D}(\mathrm{Q}_{\beta \lambda_N, N} \,\|\, \mathrm{Q}_{0,N}) \geq \limsup_{N \to \infty} \left\{ \frac{1}{2} \beta - \frac{\log M_N}{\lambda_N} \right\},$$

which, combined with Proposition 1 for some fixed $\beta > 1$, yields

$$\limsup_{N \to \infty} \frac{\lambda_N}{\log M_N} \leq 2.$$

Hence, for $\varepsilon > 0$,

$$\limsup_{N \to \infty} \frac{1}{\log M_N} \mathrm{D}(\mathrm{Q}_{2 \log M_N, N} \,\|\, \mathrm{Q}_{0,N}) \leq \limsup_{N \to \infty} \frac{1}{\log M_N} \mathrm{D}(\mathrm{Q}_{(1+\varepsilon) \lambda_N, N} \,\|\, \mathrm{Q}_{0,N})$$

$$= 2 \limsup_{N \to \infty} \frac{1}{\lambda_N} \mathrm{D}(\mathrm{Q}_{(1+\varepsilon) \lambda_N, N} \,\|\, \mathrm{Q}_{0,N})$$

$$= \varepsilon.$$

Taking $\varepsilon \to 0$ yields that $\mathrm{D}(\mathrm{Q}_{2 \log M_N, N} \,\|\, \mathrm{Q}_{0,N}) = o(\log M_N)$ and shows that we can take $\lambda_N = 2 \log M_N$. $\qquad\square$

## 4 Proof of Theorem 2: A conditional second moment method

In this section, we employ an argument known as the "conditional second moment method" to show Theorem 2. The conditional second moment method is based on two ideas: first, rather than bound the Kullback-Leilber divergence, it can be simpler to bound the $\chi^2$-divergence, which is always

an upper bound. However, the $\chi^2$ divergence can be a poor upper bound for the Kullback-Leibler divergence if the likelihood ratio has large fluctuations. The second idea is to avoid this problem by conditioning on a high-probability event which controls these fluctuations.

We make use of the following definition which is essentially borrowed from [BMV$^+$18, Section 3.3].

**Definition 2.** Write $Q_{\lambda,N}^{(\mathbf{X},\mathbf{Y})}$ for the joint distribution of $(\mathbf{X}, \mathbf{Y})$ in (1). Given a sequence $\lambda_N$, we say that a sequence of events $\Omega_N$, $N \in \mathbb{N}$ occurs with *uniformly high probability* if as $N \to +\infty$,

$$Q_{\lambda_N,N}^{(\mathbf{X},\mathbf{Y})} [\Omega_N | \mathbf{X} = x] = 1 - o(1),$$

uniformly over all $x$ in the support of $P_N$.

Given such a sequence, we write $\widetilde{Q}_{\lambda_N,N}$ for the marginal law of $\mathbf{Y}$ when we condition on the event $\Omega_N$. In the appendix, we establish the following proposition.

**Proposition 4.** *Let $\lambda_N = 2 \log M_N$. If $\Omega_N$ is a sequence of uniform high probability events, then as $N \to +\infty$, it holds*

$$D(Q_{\lambda_N,N} \,\|\, Q_{0,N}) \le D(\widetilde{Q}_{\lambda_N,N} \,\|\, Q_{0,N}) + o\left(\log M_N\right).$$

We now establish the following.

**Theorem 4.** *Assume that $\{P_N\}$ has overlap rate function $r$. Let $\lambda_N = 2 \log M_N$. There exists a uniformly high probability sequence of events $\Omega_N$ such that*

$$\limsup_{N \to \infty} \frac{1}{\lambda_N} D(\widetilde{Q}_{\lambda_N,N} \,\|\, Q_{0,N}) \le \sup_{t \in [0,1]} \left( \frac{t}{1+t} - \frac{r(t)}{2} \right)_+.$$

The hypothesis that $r(t) \ge \frac{2t}{1+t}$ implies that Assumption 1 holds, so Theorem 2 follows immediately by combining Proposition 4, Theorem 1 and Theorem 4. For the rest section we focus on proving Theorem 4.

*Proof of Theorem 4.* Let us write $\lambda_N = 2 \log M_N$. We define

$$\Omega_N = \{(x,y) : |\langle x, y \rangle - \sqrt{\lambda_N}| \le \lambda_N^{1/4}\}.$$

Since $\lambda_N \to \infty$, the sequence $\Omega_N$ occurs with uniformly high probability.

We then have

$$\frac{\mathrm{d}\widetilde{Q}_{\lambda_N,N}}{\mathrm{d}Q_{0,N}}(\mathbf{Y}) = (1 + o(1))\mathbb{E}_{\mathbf{X}} \left\{ \mathbb{1}_{\Omega_N}(\mathbf{X}, \mathbf{Y}) \cdot \exp\left( \sqrt{\lambda_N} \langle \mathbf{X}, \mathbf{Y} \rangle - \frac{\lambda_N}{2} \right) \right\},$$

which implies

$$\left( \frac{\mathrm{d}\widetilde{Q}_{\lambda_N,N}}{\mathrm{d}Q_{0,N}} \right)^2 (\mathbf{Y}) = (1+o(1))\mathbb{E}_{\mathbf{X},\mathbf{X}'} \left\{ \mathbb{1}_{\Omega_N}(\mathbf{X}, \mathbf{Y}) \mathbb{1}_{\Omega_N}(\mathbf{X}', \mathbf{Y}) \cdot \exp\left( \sqrt{\lambda_N} \langle \mathbf{X} + \mathbf{X}', \mathbf{Y} \rangle - \lambda_N \right) \right\},$$

where $\mathbf{X}, \mathbf{X}'$ are independent and identically distributed. Applying Fubini's theorem, we obtain that the chi-square divergence satisfies

$$1 + \chi^2(\widetilde{Q}_{\lambda_N,N}, Q_{0,N}) = (1 + o(1))\mathbb{E}_{\mathbf{X},\mathbf{X}'}[m_N(\mathbf{X}, \mathbf{X}')],$$

where

$$m_N(X, X') := \mathbb{E}\left\{ \mathbb{1}_{\Omega_N}(X, \mathbf{Z}) \mathbb{1}_{\Omega_N}(X', \mathbf{Z}) \cdot \exp\left( \sqrt{\lambda_N} \langle X + X', \mathbf{Z} \rangle - \lambda_N \right) \right\}, \quad \mathbf{Z} \sim Q_{0,\lambda_N}.$$

By the rotational invariance of the Gaussian measure, it is straightforward to show that $m_N$ depends only on the overlap $\langle X, X' \rangle$ between $X$ and $X'$, which we denote by $\rho$. We require the following proposition.

**Proposition 5.** *There exists a constant $C > 0$ such that for all $\rho \in [-1, 1]$, we have*

$$\frac{1}{\lambda_N} \log m_N(\rho) \le \left( \frac{\rho}{1+\rho} \right)_+ + \frac{C}{\lambda_N^{1/4}} \, ,$$

*where we define $(\rho/(1+\rho))_+ = 0$ for $\rho = -1$.*

We defer the proof of Proposition 5 to the supplement and show how this claim implies the theorem. Recall that $\mathrm{D}(\widetilde{\mathrm{Q}}_{\lambda_N, N} \, \| \, \mathrm{Q}_{0, N}) \le \log(1 + \chi^2(\widetilde{\mathrm{Q}}_{\lambda_N, N}, \mathrm{Q}_{0, N}))$. We therefore know that

$$\limsup_{N \to \infty} \frac{1}{\lambda_N} \mathrm{D}(\widetilde{\mathrm{Q}}_{\lambda_N, N} \, \| \, \mathrm{Q}_{0, N}) \le \limsup_{N \to \infty} \frac{1}{\lambda_N} \log \mathbb{E}[m_N(\rho)], \quad \rho = \langle \mathbf{X}, \mathbf{X}' \rangle \, . \tag{6}$$

We employ a standard large deviations argument. Fix a positive integer $k$. We have

$$\mathbb{E}[m_N(\rho)] \le \sum_{\ell=-k}^{k-1} \mathrm{P}_N^{\otimes 2}[\rho \ge \ell/k] \sup_{t \in [\ell/k, (\ell+1)/k)} m_N(t)$$

$$\le 2k \cdot \max_{-k \le \ell < k} \sup_{t \in [\ell/k, (\ell+1)/k)} \exp \left( \lambda_N \left( \frac{t}{1+t} \right)_+ + \log \mathrm{P}_N^{\otimes 2}[\rho \ge \ell/k] + C \lambda_N^{3/4} \right)$$

$$\le 2k \cdot \sup_{t \in [-1, 1]} \exp \left( \lambda_N \left( \frac{t}{1+t} \right)_+ + \log \mathrm{P}_N[\rho \ge t] + C \lambda_N^{3/4} + O \left( \frac{\lambda_N}{k} \right) \right) \, .$$

Since $\lambda_N \to \infty$, we have

$$\limsup_{N \to \infty} \frac{1}{\lambda_N} \log \mathbb{E}[m_N(\rho)] \le \sup_{t \in [-1, 1]} \left\{ \left( \frac{t}{1+t} \right)_+ - \frac{r(t)}{2} \right\} + O(1/k) \, ,$$

and letting $k \to \infty$ and using (6) yields

$$\limsup_{N \to \infty} \frac{1}{\lambda_N} \mathrm{D}(\widetilde{\mathrm{Q}}_{\lambda_N, N} \, \| \, \mathrm{Q}_{0, N}) \le \sup_{t \in [-1, 1]} \left\{ \left( \frac{t}{1+t} \right)_+ - \frac{r(t)}{2} \right\} \, .$$

Note that since $r$ is an overlap rate function, we must have $r(-1) = 0$; therefore, we obtain that the supremum on the right side is nonnegative. However, since the quantity in question is nonpositive when $t < 0$, we can restrict to the interval $t \in [0, 1]$ without loss of generality. This proves the claim. $\qquad\square$

## 5 Conclusion

This work shows that the all-or-nothing phenomenon in Gaussian additive models is equivalent to a condition on the Kullback-Leibler divergence between the model at a particular SNR and a standard Gaussian vector. Using this equivalence, we derive a simple condition on the overlaps which guarantees the existence of the all-or-nothing phenomenon, and as a corollary show that this phenomenon indeed arises in sparse tensor PCA for all sublinear sparsity levels.

While this paper gives a characterization of the all-or-nothing phenomenon for Gaussian models, we leave open the question of whether our framework can be extended to a more general setting. Neither the results of [RXZ19b] for sparse linear regression, nor those of [TAS20] for Bernoulli group testing, nor of [LBM20] for sparse generalized linear models are immediately implied by our main theorems. It may yet be possible to obtain more general results which encompass all of these settings.

Our results hold for the Bayes-optimal estimator $\mathbb{E}[\mathbf{X}|\mathbf{Y}]$, and we conjecture that the maximum likelihood estimator is also optimal in this setting and thereby also exhibits the all-or-nothing phenomenon. However, neither of these estimators is computationally efficient in general. Interestingly, in [RXZ19a] the authors study, in the context of sparse linear regression, the all-or-nothing phenomenon for the performance of a well-studied computationally efficient algorithm called Approximate Message Passing. Developing a general theory for the optimal recovery thresholds for certain families of polynomial-time estimators—and establishing whether similar all-or-nothing behavior holds—is an important question for future work.

## Broader impact

This work does not present any questions of immediate societal impact.

## Acknowledgments and Disclosure of Funding

JNW acknowledges the support of the Institute for Advanced Study, where a portion of this research was conducted. IZ is supported by a CDS Moore-Sloan postdoctoral fellowship.

## Footnotes

[1]We denote the dimension by $p$ rather than $N$ in anticipation of the generalization which will appear in the following section.

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
