[Supplementary Material]

## A Omitted proofs

*Proof of Proposition 1.* We recall the I-MMSE relation (11):

$$\frac{d}{d\beta}\frac{1}{\lambda_N}\mathrm{D}(\mathrm{Q}_{\beta\lambda_N,N}\|\mathrm{Q}_{0,N}) = \frac{1}{2} - \frac{1}{2}\mathrm{MMSE}_N(\beta\lambda_N)\,.$$

Let us first assume that the all-or-nothing phenomenon holds. Since $\mathrm{D}(\mathrm{Q}_{0,N}\|\mathrm{Q}_{0,N}) = 0$, we can write

$$\begin{aligned}
\lim_{N\to\infty}\frac{1}{\lambda_N}\mathrm{D}(\mathrm{Q}_{\beta\lambda_N,N}\|\mathrm{Q}_{0,N}) &= \lim_{N\to\infty}\int_0^\beta \frac{d}{d\kappa}\frac{1}{\lambda_N}\mathrm{D}(\mathrm{Q}_{\kappa\lambda_N,N}\|\mathrm{Q}_{0,N})\,d\kappa \\
&= \lim_{N\to\infty}\int_0^\beta \frac{1}{2} - \frac{1}{2}\mathrm{MMSE}_N(\kappa\lambda_N)\,d\kappa \\
&\overset{(*)}{=} \int_0^\beta \frac{1}{2} - \lim_{N\to\infty}\mathrm{MMSE}_N(\kappa\lambda_N)\,d\kappa \\
&= \frac{1}{2}(\beta-1)_+\,,
\end{aligned}$$

where in $(*)$ we have used the dominated convergence theorem and the fact that $\mathrm{MMSE}_N(\kappa\lambda_N) \in [0,1]$ and where the last equality follows from the all-or-nothing phenomenon.

In the other direction, we use the fact that $\mathrm{MMSE}_N(\beta\lambda_N)$ is a non-increasing function of $\beta$ [see, e.g., Mio19, Proposition 1.3.1]. Combined with the I-MMSE relation, this immediately yields that $\frac{1}{\lambda_N}\mathrm{D}(\mathrm{Q}_{\beta\lambda_N,N}\|\mathrm{Q}_{0,N})$ is convex. We therefore have by standard facts in convex analysis [HUL93, Proposition 4.3.4] that

$$\frac{1}{2} - \frac{1}{2}\lim_{N\to\infty}\mathrm{MMSE}_N(\beta\lambda_N) = \lim_{N\to\infty}\frac{d}{d\beta}\frac{1}{\lambda_N}\mathrm{D}(\mathrm{Q}_{\beta\lambda_N,N}\|\mathrm{Q}_{0,N}) = \frac{d}{d\beta}\left(\lim_{N\to\infty}\frac{1}{\lambda_N}\mathrm{D}(\mathrm{Q}_{\beta\lambda_N,N}\|\mathrm{Q}_{0,N})\right)$$

for all $\beta$ for which the right side exists. Since we have assumed that

$$\lim_{N\to\infty}\frac{1}{\lambda_N}\mathrm{D}(\mathrm{Q}_{\beta\lambda_N,N}\|\mathrm{Q}_{0,N}) = \frac{1}{2}(\beta-1)_+,$$

the right side is 0 when $\beta < 1$ and $\frac{1}{2}$ when $\beta > 1$. The all-or-nothing property immediately follows. $\square$

*Proof of Proposition 2.* The first claim follows directly from Lemma 6. Indeed, for the sparse vector model, $\log M_p = (1+o(1))k\log\frac{p}{k}$, and by Lemma 6,

$$\lim \frac{1}{k\log\frac{p}{k}}\log \mathrm{P}_p^{\otimes 2}[\langle \mathbf{X},\mathbf{X}'\rangle \geq t] = -t\,. \tag{7}$$

Since $t < \frac{2t}{1+t}$ for all $t\in(0,1)$, the claim holds.

We now turn to the proof of the all-or-nothing phenomenon. By Theorem 1, it suffices to show

$$\mathrm{D}(\mathrm{Q}_{2\log M_p,p}\|\mathrm{Q}_{0,p}) = o\left(\log M_p\right)\,.$$

We write

$$\begin{aligned}
\mathrm{D}(\mathrm{Q}_{2\log M_p,p}\|\mathrm{Q}_{0,p}) &= \mathbb{E}_{\mathbf{Y}\sim\mathrm{Q}_{2\log M_p,p}}\log\mathbb{E}_{\mathbf{X}'\sim\mathrm{P}_p}\exp\left(\sqrt{2\log M_p}\langle\mathbf{Y},\mathbf{X}'\rangle - \log M_p\right) \\
&= \mathbb{E}_{\mathbf{X}}\mathbb{E}_{\mathbf{Z}}\log\mathbb{E}_{\mathbf{X}'\sim\mathrm{P}_p}\exp\left(\sqrt{2\log M_p}\langle\mathbf{Z},\mathbf{X}'\rangle + 2\log M_p\langle\mathbf{X},\mathbf{X}'\rangle - \log M_p\right)
\end{aligned}$$

Now, given $\mathbf{X}$ and any vector $v\in\mathbb{R}^p$, let us denote by $v|_{\mathbf{X}}\in\mathbb{R}^p$ the vector given by

$$(v|_{\mathbf{x}})_i := \begin{cases} v_i & \text{if } \mathbf{X}_i \neq 0, \\ 0 & \text{otherwise.} \end{cases}$$

Similarly, let $v|_{\mathbf{X}^c} := v - v|_{\mathbf{X}}$. Given $\mathbf{X}$, the vectors $\mathbf{Z}|_{\mathbf{X}}$ and $\mathbf{Z}|_{\mathbf{X}^c}$ are independent; thus we can apply Jensen's inequality to the expectation with respect to $\mathbf{Z}|_{\mathbf{X}^c}$ to obtain

$$D(Q_{2\log M_p, p} \,\|\, Q_{0,p}) \le \mathbb{E}_{\mathbf{X}} \mathbb{E}_{\mathbf{Z}|_{\mathbf{X}}} \log \mathbb{E}_{\mathbf{X}' \sim P_p} \mathbb{E}_{\mathbf{Z}|_{\mathbf{X}^c}} \exp\left( \sqrt{2\log M_p} \langle \mathbf{Z}, \mathbf{X}' \rangle + 2\log M_p \langle \mathbf{X}, \mathbf{X}' \rangle - \log M_p \right)$$

$$= \mathbb{E}_{\mathbf{X}} \mathbb{E}_{\mathbf{Z}|_{\mathbf{X}}} \log \mathbb{E}_{\mathbf{X}' \sim P_p} \exp\left( \sqrt{2\log M_p} \langle \mathbf{Z}|_{\mathbf{X}}, \mathbf{X}'|_{\mathbf{X}} \rangle + \log M_p (\|\mathbf{X}'|_{\mathbf{X}^c}\|^2 + 2\langle \mathbf{X}, \mathbf{X}' \rangle - 1) \right) .$$

Since the entries of $\mathbf{X}$ and $\mathbf{X}'$ are all either $0$ or $1/\sqrt{k}$ and $\mathbf{X}$ has unit norm, we have that $\langle \mathbf{X}, \mathbf{X}' \rangle = \|\mathbf{X}'|_{\mathbf{X}}\|^2$, and since $\mathbf{X}'|_{\mathbf{X}^c}$ and $\mathbf{X}'|_{\mathbf{X}}$ are orthogonal, we obtain

$$\|\mathbf{X}'|_{\mathbf{X}^c}\|^2 + 2\langle \mathbf{X}, \mathbf{X}' \rangle - 1 = \langle \mathbf{X}, \mathbf{X}' \rangle .$$

Continuing from above and using that $\|\mathbf{X}'|_{\mathbf{X}}\|_\infty \le 1/\sqrt{k}$, we have

$$D(Q_{2\log M_p, p} \,\|\, Q_{0,p}) \le \mathbb{E}_{\mathbf{X}} \mathbb{E}_{\mathbf{Z}|_{\mathbf{X}}} \log \mathbb{E}_{\mathbf{X}' \sim P_p} \exp\left( \sqrt{2\log M_p/k} \|\mathbf{Z}|_{\mathbf{X}}\|_1 + \log M_p \langle \mathbf{X}, \mathbf{X}' \rangle \right)$$

$$= \mathbb{E}_{\mathbf{X}} \log \mathbb{E}_{\mathbf{X}' \sim P_p} \exp\left( \log M_p \langle \mathbf{X}, \mathbf{X}' \rangle \right) + \mathbb{E}_{\mathbf{X}} \mathbb{E}_{\mathbf{Z}|_{\mathbf{X}}} \sqrt{(2\log M_p/k)} \|\mathbf{Z}|_{\mathbf{X}}\|_1$$

$$\le \mathbb{E}_{\mathbf{X}} \log \mathbb{E}_{\mathbf{X}' \sim P_p} \exp\left( \log M_p \langle \mathbf{X}, \mathbf{X}' \rangle \right) + O\left( \sqrt{2k \log M_p} \right)$$

Since $k = o(p)$, we also have that $k = o(k \log \frac{p}{k}) = o(\log M_p)$; therefore, the second term is $o(\log M_p)$. Hence it suffices to focus on the first term.

We proceed via a large deviations argument as in the proof of Theorem 4. Write $\rho = \langle \mathbf{X}, \mathbf{X}' \rangle$ for the overlap; note that the law of $\rho$ is the same for all $\mathbf{X}$ in the support of $P_p$, so it suffices to understand $\log \mathbb{E} \exp(\rho \log M_p)$. We have, for any fixed positive integer $\ell$,

$$\mathbb{E} \exp(\rho \log M_p) \le \sum_{m=0}^{\ell-1} P_N[\rho \ge m/\ell] \exp\left( \frac{m+1}{\ell} \log M_p \right)$$

$$\le \ell \cdot \max_{0 \le m < \ell} \exp\left( \frac{m+1}{\ell} \log M_p + \log P_N[\rho \ge m/\ell] \right) ,$$

which implies

$$\limsup_{p \to \infty} \frac{1}{\log M_p} \log \mathbb{E} \exp(\rho \log M_p) \le \max_{0 \le m < \ell} \frac{m+1}{\ell} - \frac{m}{\ell} ,$$

where we have used (7). Therefore $\limsup_{p \to \infty} \frac{1}{\log M_p} \log \mathbb{E} \exp(\rho \log M_p) = O(1/\ell)$, and letting $\ell \to \infty$ proves the claim. □

*Proof of Proposition 3.* Denote by $\mathcal{S}_k$ the set of $k$-sparse vectors in $\mathbb{R}^p$. Note that the cardinality of $\{0, 1/\sqrt{k}\}^p \cap \mathcal{S}_k$ is $\binom{p}{k}$ and the cardinality of $\{-1/\sqrt{k}, 0, 1/\sqrt{k}\}^p \cap \mathcal{S}_k$ is $\binom{p}{k} 2^k$. In the case of the Bernoulli prior, the identification $\mathbf{x} \mapsto x^{\otimes d}$ is a bijection, so $M_N$ for the Bernoulli prior is $\binom{p}{k}$. In the case of the Bernoulli-Rademacher prior, when $d$ is odd the map $\mathbf{x} \mapsto x^{\otimes d}$ is still a bijection, but when $d$ is even, the vectors $\mathbf{x}$ and $-\mathbf{x}$ give rise to the same tensor. Therefore $M_N$ for the Bernoulli-Rademacher prior is either $\binom{p}{k} 2^k$ or $\binom{p}{k} 2^{k-1}$. Nevertheless, using Stirling's approximation, since $k = o(p)$, we have for both the Bernoulli and Bernoulli-Rademacher prior that

$$\log M_N = (1 + o(1)) k \log \frac{p}{k} .$$

Now notice that the overlap $\langle \mathbf{X}, \mathbf{X}' \rangle$ in the case that $\mathbf{x}$ is Bernoulli-Rademacher is stochastically dominated by the overlap when $\mathbf{x}$ is Bernoulli. To prove this, let us consider the natural coupling between the two different priors on $\mathbf{x}$: we first sample $\mathbf{x}_1$ from the sparse Bernoulli distribution and then choose uniformly at random the signs for the non-zero values of $\mathbf{x}_1$ to form a sample $\mathbf{x}_2$ from the Bernoulli-Rademacher distribution. Notice that by triangle inequality under this coupling it holds almost surely

$$\langle \mathbf{x}_2^{\otimes d}, \mathbf{x}_2'^{\otimes d} \rangle \le |\langle \mathbf{x}_2^{\otimes d}, \mathbf{x}_2'^{\otimes d} \rangle| \le \langle \mathbf{x}_1^{\otimes d}, \mathbf{x}_1'^{\otimes d} \rangle.$$

For this reason it suffices to prove our result only in the case the prior $\widetilde{P}_p$ is the uniform distribution over $\{0, 1/\sqrt{k}\}^p \cap \mathcal{S}_k$. We therefore focus on this case in the rest of the proof.

Now fix any $t \in [0, 1]$ and notice that by elementary algebra for any $v, v' \in \mathbb{R}^p$ with $\|v\| = \|v'\| = 1$ since $d \geq 2$ it holds $\langle v^{\otimes d}, v'^{\otimes d} \rangle = \langle v, v' \rangle^d \leq \langle v, v' \rangle^2$. Hence as $\mathbf{x}, \mathbf{x}'$ live on the sphere of dimension $p$,

$$\begin{aligned}
\mathrm{P}_N^{\otimes 2}[\langle \mathbf{X}, \mathbf{X}' \rangle \geq t] = \widetilde{\mathrm{P}}_p^{\otimes 2}[\langle \mathbf{x}^{\otimes d}, \mathbf{x}'^{\otimes d} \rangle \geq t] &= \widetilde{\mathrm{P}}_p^{\otimes 2}[\langle \mathbf{x}, \mathbf{x}' \rangle^d \geq t] \\
&\leq \widetilde{\mathrm{P}}_p^{\otimes 2}[\langle \mathbf{x}, \mathbf{x}' \rangle^2 \geq t] \\
&= \widetilde{\mathrm{P}}_p^{\otimes 2}[\langle \mathbf{x}, \mathbf{x}' \rangle \geq \sqrt{t}].
\end{aligned} \tag{8}$$

Since $\mathbf{x}, \mathbf{x}'$ are drawn from the uniform distribution over $\{0, 1/\sqrt{k}\}^p \cap \mathcal{S}_k$, Lemma 6 combined with (8) yields

$$\lim_{N \to +\infty} \frac{1}{\log M_N} \log \mathrm{P}_N^{\otimes 2}[\langle \mathbf{X}, \mathbf{X}' \rangle \geq t] \leq -\sqrt{t}.$$

The elementary inequality $-\sqrt{t} \leq -\frac{2t}{1+t}$ concludes the proof. $\qquad\square$

*Proof of Proposition 4.* Let

$$Z(Y) = \frac{\mathrm{Q}_{\lambda_N, N}(Y)}{\mathrm{Q}_{0,N}(Y)} = \mathbb{E}_{\mathbf{X}' \sim \mathrm{P}_N} \exp\left( \sqrt{\lambda_N} \langle Y, \mathbf{X}' \rangle - \frac{\lambda_N}{2} \right)$$

Following *mutatis mutandis* the first two arguments in the proof of [BMV+18, Theorem 5] we obtain

$$\mathrm{D}(\mathrm{Q}_{\lambda_N, N} \| \mathrm{Q}_{0,N}) \leq \mathrm{D}(\widetilde{\mathrm{Q}}_{\lambda_N, N} \| \mathrm{Q}_{0,N}) + o(1) \cdot \sqrt{\mathbb{E}_{\mathbf{Y} \sim \mathrm{Q}_{\lambda_N, N}} \left[ \log^2 Z(\mathbf{Y}) \right]}. \tag{9}$$

It is straightforward to see that for all $Y$,

$$|\log Z(Y)| \leq \sqrt{\lambda_N} \max_{X' \in \mathrm{Support}(P_N)} \langle X', Y \rangle + \frac{\lambda_N}{2}$$

which implies that

$$\mathbb{E}_{\mathbf{Y} \sim \mathrm{Q}_{\lambda_N, N}} \log^2 Z(\mathbf{Y}) \leq 2\lambda_N \cdot \mathbb{E}_{\mathbf{Y} \sim \mathrm{Q}_{\lambda_N, N}} \max_{X' \in \mathrm{Support}(P_N)} \langle X', \mathbf{Y} \rangle^2 + O\left(\lambda_N^2\right). \tag{10}$$

Now recall $\mathbf{Y} = \sqrt{\lambda_N} \mathbf{X} + \mathbf{Z}$ for $\mathbf{Z} \sim Q_{0,N}$ and for all $X' \in \mathrm{Support}(P_N)$ it holds $|\langle \mathbf{X}, X' \rangle| \leq \|\mathbf{X}\| \|X'\| = 1$ almost surely. Hence,

$$\begin{aligned}
\mathbb{E}_{\mathbf{Y} \sim \mathrm{Q}_{\lambda_N, N}} \max_{X' \in \mathrm{Support}(P_N)} \langle X', \mathbf{Y} \rangle^2 &= \mathbb{E}_{\mathbf{Z} \sim Q_{0,N}} \left( \max_{X' \in \mathrm{Support}(P_N)} |\sqrt{\lambda_N} \langle X', \mathbf{X} \rangle + \langle X', \mathbf{Z} \rangle| \right)^2 \\
&\leq 2\lambda_N + 2\mathbb{E}_{\mathbf{Z} \sim Q_{0,N}} \max_{X' \in \mathrm{Support}(P_N)} \langle X', \mathbf{Z} \rangle^2.
\end{aligned}$$

Since $\mathbf{Q}_{0,N}$ is simply the law of a vector with i.i.d. standard Gaussian coordinates and the cardinality of the discrete subset of the sphere $\mathrm{Support}(P_N)$ is equal to $M_N$, by Lemma 5 we have $\mathbb{E}_{\mathbf{Z} \sim Q_{0,N}} \max_{X' \in \mathrm{Support}(P_N)} \langle X', \mathbf{Z} \rangle^2 = O(\log M_N)$. Therefore since $\lambda_N = O(\log M_N)$,

$$\mathbb{E}_{\mathbf{Y} \sim Q_{\lambda_N, N}} \max_{X' \in \mathrm{Support}(P_N)} \langle X', \mathbf{Y} \rangle^2 \leq O(\lambda_N + \log M_N) = O(\log M_N).$$

Combining the last inequality with (10), we conclude that

$$\mathbb{E}_{\mathbf{Y} \sim Q_{\lambda_N, N}} \log^2 Z(\mathbf{Y}) = O\left(\lambda_N^2\right) = O\left(\log^2 M_N\right).$$

Using (9) completes the proof of the proposition. $\qquad\square$

*Proof of Proposition 5.* We let $C$ denote an absolute positive constant whose value may change from line to line. Let us write $\mathbf{W} = \langle X, \mathbf{Z} \rangle / \sqrt{\lambda_N}$ and $\mathbf{W}' = \langle X', \mathbf{Z} \rangle / \sqrt{\lambda_N}$. Recall that $X, X'$ lie on the unit sphere with $\langle X, X' \rangle = \rho$.

Then $\mathbf{W}$ and $\mathbf{W}'$ are are jointly Gaussian with mean $0$ and covariance $\frac{1}{\lambda_N}\begin{pmatrix} 1 & \rho \\ \rho & 1 \end{pmatrix} =: \frac{1}{\lambda_N}\Sigma_\rho$. Under this parametrization, we have

$$\exp(\sqrt{\lambda_N}(\langle X, \mathbf{Z}\rangle + \langle X', \mathbf{Z}\rangle) - \lambda_N) = \exp(\lambda_N(\mathbf{W} + \mathbf{W}' - 1)).$$

Let us write $S$ for the set $\{(w, w') : |w - 1| \le \lambda_N^{-1/4}, |w' - 1| \le \lambda_N^{-1/4}\}$.

We consider three cases:

**Case 1:** $\rho \le 0$   Using the moment generating function of the univariate normal distribution yields

$$\mathbb{E}\exp(\lambda_N(\mathbf{W} + \mathbf{W}' - 1))\mathbb{1}_S(\mathbf{W}, \mathbf{W}') \le \mathbb{E}\exp(\lambda_N(\mathbf{W} + \mathbf{W}' - 1)) = e^{\lambda_N \rho} \le 1,$$

so

$$\frac{1}{\lambda_N}\log m_N(\rho) \le 0 = \left(\frac{\rho}{1 + \rho}\right)_+ .$$

**Case 2:** $\rho \in (0, 1/2]$   Write $\phi_\rho(w, w')$ for the joint density of $\mathbf{W}$ and $\mathbf{W}'$. Note that on $S$

$$\phi_\rho(w, w') \le \frac{\lambda_N}{2\pi(1 - \rho^2)}\exp\left(-\frac{\lambda_N}{2}\mathbf{w}^\top \Sigma_\rho^{-1} \mathbf{w}\right), \qquad \mathbf{w} = (w, w')$$

$$\le Ce^{-\frac{\lambda_N}{1+\rho} + C\lambda_N^{3/4}},$$

where we use that $\lambda_N \to +\infty$ as $N \to +\infty$. Hence

$$\frac{1}{\lambda_N}\log m_N(\rho) = \frac{1}{\lambda_N}\log\int_S e^{\lambda_N(w + w' - 1)}\phi_\rho(w, w')\,\mathrm{d}w\,\mathrm{d}w'$$

$$\le \frac{1}{\lambda_N}\log\int_S \max_{(w,w')\in S} e^{\lambda_N(w + w' - 1)} \cdot \max_{(w,w')\in S}\phi_\rho(w, w')\,\mathrm{d}w\,\mathrm{d}w'$$

$$\le \frac{1}{\lambda_N}\log(\mathrm{vol}(S) \cdot e^{\lambda_N + O(\lambda_N^{3/4})} \cdot Ce^{-\frac{\lambda_N}{1+\rho} + C\lambda_N^{3/4}})$$

$$\le \frac{\rho}{1 + \rho} + \frac{C}{\lambda_N^{1/4}} .$$

**Case 3:** $\rho \in (1/2, 1]$   The sum $\mathbf{W} + \mathbf{W}'$ is Gaussian with mean $0$ and variance $\frac{2}{\lambda_N}(1 + \rho)$, and if $(w, w') \in S$, then $|w + w' - 2| \le 2\lambda_N^{-1/4}$.

We obtain

$$m_N(\rho) = \mathbb{E}\exp(\lambda_N(\mathbf{W} + \mathbf{W}' - 1))\mathbb{1}_S(\mathbf{W}, \mathbf{W}') \le \mathbb{E}\exp(\lambda_N(\mathbf{W}'' - 1))\mathbb{1}_{|\mathbf{W}'' - 2| \le 2\lambda_N^{-1/4}},$$

where $\mathbf{W}'' \sim \mathcal{N}(0, \frac{2}{\lambda_N}(1 + \rho))$. Similar with the analysis in Case 2, the density of $\mathbf{W}''$ is bounded by $Ce^{-\frac{\lambda_N}{1+\rho} + C\lambda_N^{3/4}}$ on the set $T := \{w'' : |w'' - 2| \le 2\lambda_N^{-1/4}\}$, and we obtain

$$\frac{1}{\lambda_N}\log m_N(\rho) \le \frac{1}{\lambda_N}\log\int_T \max_{w''\in T} e^{\lambda_N(w'' - 1)} \cdot Ce^{-\frac{\lambda_N}{1+\rho} + C\lambda_N^{3/4}}$$

$$\le \frac{1}{\lambda_N}\log(\mathrm{vol}(T) \cdot e^{\lambda_N + O(\lambda_N^{3/4})} \cdot Ce^{-\frac{\lambda_N}{1+\rho} + C\lambda_N^{3/4}})$$

$$\le \frac{\rho}{1 + \rho} + \frac{C}{\lambda_N^{1/4}} ,$$

as claimed. □

## B   Additional lemmas

**Lemma 1.** *For all $N$ and $\lambda > 0$, the function $\beta \mapsto \frac{1}{\lambda}\mathrm{D}(\mathrm{Q}_{\beta\lambda, N} \,\|\, \mathrm{Q}_{0, N})$ is nonnegative, nondecreasing, and $1/2$-Lipschitz.*

*Proof.* Let us fix some $N$ and $\lambda$. The nonnegativity follows from the nonnegativity of the KL divergence. By Lemma 2, we have

$$\frac{1}{\lambda}\,\mathrm{D}(\mathrm{Q}_{\beta\lambda,N}\,\|\,\mathrm{Q}_{0,N}) = \frac{\beta}{2} - \frac{1}{\lambda}I_{\beta\lambda,N}(\mathbf{X};\mathbf{Y})\,.$$

Differentiating with respect to $\beta$ and using the I-MMSE theorem [GSV05] we conclude

$$\frac{d}{d\beta}\frac{1}{\lambda}\,\mathrm{D}(\mathrm{Q}_{\beta\lambda,N}\,\|\,\mathrm{Q}_{0,N}) = \frac{1}{2} - \frac{1}{2}\,\mathrm{MMSE}_N(\beta\lambda). \tag{11}$$

The results that $\beta \mapsto \frac{1}{\lambda}\mathrm{D}(\mathbf{Y}_{\beta\lambda}\,\|\,\mathbf{Z})$ is nondecreasing and $1/2$-Lipschitz follow directly from the fact that $\mathrm{MMSE}_N(\beta\lambda) \in [0,1]$. $\qquad\square$

**Lemma 2.** *Denote by $I_{\lambda,N}(\mathbf{X};\mathbf{Y})$ the mutual information between $\mathbf{X}$ and $\mathbf{Y}$ in (1), and denote by $\mathrm{Q}_{\lambda,N}^{(\mathbf{X},\mathbf{Y})}$ their joint law. Then*

$$I_{\lambda,N}(\mathbf{X};\mathbf{Y}) = \mathrm{D}(\mathrm{Q}_{\lambda,N}^{(\mathbf{X},\mathbf{Y})}\,\|\,\mathrm{P}_N \otimes \mathrm{Q}_{\lambda,N}) = \frac{\lambda}{2} - \mathrm{D}(\mathrm{Q}_{\lambda,N}\,\|\,\mathrm{Q}_{0,N})\,.$$

*Proof.* The first equality is the definition of mutual information. We then have

$$\mathrm{D}(\mathrm{Q}_{\lambda,N}^{(\mathbf{X},\mathbf{Y})}\,\|\,\mathrm{P}_N \otimes \mathrm{Q}_{\lambda,N}) = \mathbb{E}_{\mathrm{Q}_{\lambda,N}^{(\mathbf{x},\mathbf{Y})}} \log \frac{\mathrm{Q}_{\lambda,N}(\mathbf{Y}|\mathbf{X})}{\mathrm{Q}_{\lambda,N}(\mathbf{Y})}$$

$$= \mathbb{E}_{\mathrm{Q}_{\lambda,N}^{(\mathbf{x},\mathbf{Y})}} \log \frac{\mathrm{Q}_{\lambda,N}(\mathbf{Y}|\mathbf{X})}{\mathrm{Q}_{0,N}(\mathbf{Y})} - \mathbb{E}_{\mathrm{Q}_{\lambda,N}} \log \frac{\mathrm{Q}_{\lambda,N}(\mathbf{Y})}{\mathrm{Q}_{0,N}(\mathbf{Y})}\,.$$

Using the fact that $\mathbf{Z}$ has i.i.d. standard Gaussian entries we have

$$\mathbb{E}_{\mathrm{Q}_{\lambda,N}^{(\mathbf{x},\mathbf{Y})}} \log \frac{\mathrm{Q}_{\lambda,N}(\mathbf{Y}|\mathbf{X})}{\mathrm{Q}_0(\mathbf{Y})} = \mathbb{E}_{\mathrm{Q}_{\lambda,N}^{(\mathbf{x},\mathbf{Y})}} \frac{\|\mathbf{Y}\|_2^2 - \|\mathbf{Y} - \sqrt{\lambda}\mathbf{X}\|_2^2}{2} = \frac{\lambda}{2}\,,$$

and by definition

$$\mathrm{D}(\mathrm{Q}_{\lambda,N}\,\|\,\mathrm{Q}_{0,N}) = \mathbb{E}_{\mathrm{Q}_{\lambda,N}} \log \frac{\mathrm{Q}_{\lambda,N}(\mathbf{Y})}{\mathrm{Q}_{0,N}(\mathbf{Y})}.$$

The claim follows. $\qquad\square$

**Lemma 3.** *For all $\lambda \geq 0$,*

$$\mathrm{D}(\mathrm{Q}_{\lambda,N}\,\|\,\mathrm{Q}_{0,N}) \geq \frac{\lambda}{2} - \log M_N\,.$$

*Proof.* Writing explicitly the Kullback-Leibler divergence gives

$$\mathrm{D}(\mathrm{Q}_{\lambda,N}\,\|\,\mathrm{Q}_{0,N}) = \mathbb{E}\log \frac{1}{M_N}\sum_{X'\in\mathrm{Support}(\mathrm{P}_N)} \exp\left(\sqrt{\lambda}\langle\mathbf{Y},X'\rangle - \frac{\lambda}{2}\right) \qquad \mathbf{Y}\sim\mathrm{Q}_{\lambda,N}$$

$$\geq \mathbb{E}\log \frac{1}{M_N}\exp\left(\sqrt{\lambda}\langle\mathbf{Z},\mathbf{X}\rangle + \frac{\lambda}{2}\right)$$

$$= \mathbb{E}\left\{\sqrt{\lambda}\langle\mathbf{Z},\mathbf{X}\rangle + \frac{\lambda}{2} - \log M_N\right\} = \frac{\lambda}{2} - \log M_N\,,$$

where the inequality follows from writing $\mathbf{Y} = \sqrt{\lambda}\mathbf{X} + \mathbf{Z}$ and taking only the $X' = \mathbf{X}$ term in the sum. $\qquad\square$

**Lemma 4.** *Let $\alpha_1 = (\alpha_1)_{N\in\mathbb{N}}$ and $\alpha_2 = (\alpha_2)_{N\in\mathbb{N}}$ be two sequences in $[0,1]$ such that $\alpha_1 = 1-o(1)$ and $\alpha_2 = o(1)$ as $N \to \infty$, and let $\lambda_N$ be any sequence tending to infinity as $N \to +\infty$ such that $\frac{1}{\lambda_N}d(\alpha_1\,\|\,\alpha_2)$ is bounded. Then*

$$\limsup_{N\to\infty} \frac{1}{\lambda_N}d(\alpha_1\,\|\,\alpha_2) = \limsup_{N\to\infty} \frac{1}{\lambda_N}\log\frac{1}{\alpha_2}\,.$$

*Proof.* The given asymptotics imply

$$\lim_{N\to\infty}(1-\alpha_1)\log\frac{1-\alpha_1}{1-\alpha_2}=0\,.$$

Moreover, since $\alpha_1\log\alpha_1$ is bounded, we have

$$\lim_{N\to\infty}\frac{1}{\lambda_N}\alpha_1\log\alpha_1=0\,.$$

Combining these facts yields

$$\limsup_{N\to\infty}\frac{1}{\lambda_N}d(\alpha_1\,\|\,\alpha_2)=\limsup_{N\to\infty}\frac{1}{\lambda_N}\alpha_1\log\frac{\alpha_1}{\alpha_2}+(1-\alpha_1)\log\frac{1-\alpha_1}{1-\alpha_2}$$
$$=\limsup_{N\to\infty}\frac{1}{\lambda_N}\alpha_1\log\frac{1}{\alpha_2}\,.$$

Since $\frac{1}{\lambda_N}d(\alpha_1\,\|\,\alpha_2)$ is bounded, so is the sequence $\frac{1}{\lambda_N}\alpha_1\log\frac{1}{\alpha_2}$, and since $\alpha_1$ is bounded away from 0, this implies that $\frac{1}{\lambda_N}\log\frac{1}{\alpha_2}$ is bounded as well. Using that $\lim_{N\to\infty}\alpha_1=1$ therefore yields the claim. □

**Lemma 5.** *Let $M,N\in\mathbb{N}$ and let $S$ be a discrete subset of the $N$-dimensional unit sphere with cardinality $M$. Then for $G$ the law of the $N$-dimensional random variable $\mathbf{Z}$ with i.i.d. standard Gaussian coordinates it holds*

$$E_{\mathbf{Z}\sim G}\max_{X'\in S}\langle X',\mathbf{Z}\rangle^2=O\left(\log M\right).$$

*Proof.* It suffices to show that

$$E_{\mathbf{Z}\sim G}\max_{X'\in S}\langle X',\mathbf{Z}\rangle^2\mathbb{1}\left(\max_{X'\in S}\langle X',\mathbf{Z}\rangle^2\geq 2\log M\right)=O\left(1\right).$$

or

$$\int_0^\infty G\left(\max_{X'\in S}\langle X',\mathbf{Z}\rangle^2\geq 2\log M+t\right)\mathrm{d}t=O\left(1\right).$$

Using a union bound argument and the fact that for all $X'\in S$ the quantity $\langle X',\mathbf{Z}\rangle$ follows a standard Gaussian distribution, we have for all $t\geq 0$,

$$G\left(\max_{X'\in S}\langle X',\mathbf{Z}\rangle^2\geq 2\log M+t\right)\leq M\exp\left(-\log M-\frac{t}{2}\right)=\exp\left(-\frac{t}{2}\right).$$

Hence

$$\int_0^\infty G\left(\max_{X'\in S}\langle X',\mathbf{Z}\rangle^2\geq 2\log M+t\right)\mathrm{d}t\leq\int_0^\infty\exp\left(-\frac{t}{2}\right)\mathrm{d}t=O\left(1\right),$$

as we wanted.

□

**Lemma 6.** *Suppose that $k=o(p)$ and the prior $\widetilde{P}_p$ is the uniform distribution on all the $k$-sparse vectors with elements either $0$ or $1/\sqrt{k}$. Then for any $t\in[0,1]$ it holds*

$$\lim_{p\to+\infty}\frac{1}{k\log\frac{p}{k}}\log\widetilde{P}_p^{\otimes 2}[\langle\mathbf{x},\mathbf{x}'\rangle\geq t]=-t.$$

*Proof.* First note that the claim follows immediately when $t=1$ as when $k=o(p)$, the distribution $\widetilde{P}_p$ is distribution over a discrete subset of the unit sphere of cardinality $(1+o(1))k\log\frac{p}{k}$. Similarly, since for all $v,v'$ in the support of $\widetilde{P}_p$ it holds $\langle v,v'\rangle\geq 0$, the claim also follows straightforwardly for $t=0$. For the rest of the proof we assume $t\in(0,1)$.

We first show that the limit superior is bounded above by $-t$. The distribution of the rescaled overlap $k\langle \mathbf{x}, \mathbf{x}'\rangle = \langle \sqrt{k}\mathbf{x}, \sqrt{k}\mathbf{x}'\rangle$ follows the Hypergeometric distribution $\mathrm{Hyp}\,(p, k, k)$ with probability mass function $p(s) = \binom{k}{s}\binom{p-k}{k-s}/\binom{p}{k}$, for $s = 0, \ldots, k$. Therefore for a fixed $t \in (0, 1]$,

$$\widetilde{\mathrm{P}}_p^{\otimes 2}[\langle \mathbf{x}, \mathbf{x}'\rangle \geq t] = \sum_{s=\lceil tk \rceil}^{k} p(s). \tag{12}$$

Now for any $s \geq \lceil tk \rceil$ it holds

$$\frac{p(s+1)}{p(s)} = \frac{\binom{k}{s+1}}{\binom{k}{s}}\frac{\binom{p-k}{k-s-1}}{\binom{p-k}{k-s}} = \frac{(k-s)^2}{(s+1)(p-2k+s+1)}.$$

Using that $k = o(p)$ and $s \geq tk$ we conclude that for sufficiently large $p$ and all $s \geq \lceil tk \rceil$ it holds

$$\frac{p(s+1)}{p(s)} \leq 2\frac{k}{tp} < \frac{1}{2}.$$

or by telescopic product,

$$\frac{p(s)}{p(\lceil tk \rceil)} \leq \frac{1}{2^{s-\lceil tk \rceil}}.$$

Hence, using (12) we have for large enough values of $p$,

$$\widetilde{\mathrm{P}}_p^{\otimes 2}[\langle \mathbf{x}, \mathbf{x}'\rangle \geq t] \leq \sum_{s=\lceil tk \rceil}^{k} p(\lceil tk \rceil)\frac{1}{2^{s-\lceil tk \rceil}} \leq 2p(\lceil tk \rceil). \tag{13}$$

We have

$$p(\lceil tk \rceil) = \binom{k}{\lceil tk \rceil}\binom{p-k}{k-\lceil tk \rceil}/\binom{p}{k}$$

and combining with the elementary bound $\log \binom{m_1}{m_2} = m_2 \log \left(\frac{em_1}{m_2}\right) + O(m_2)$, for $m_1 \leq m_k$, we obtain

$$\log p(\lceil tk \rceil) = tk \log \frac{1}{t} + (1-t)k \log \frac{p-k}{(1-t)k} - k \log \frac{p}{k} + O(k)$$

$$= -tk \log \frac{p}{k} + O(k), \tag{14}$$

where in the second step we have used that, for fixed $t \in (0, 1)$, if $k = o(p)$, then

$$\log \frac{p-k}{(1-t)k} = \log \frac{p}{k} + O(1).$$

We therefore conclude

$$\log \widetilde{\mathrm{P}}_p^{\otimes 2}[\langle \mathbf{x}, \mathbf{x}'\rangle \geq t] \leq \log p(\lceil tk \rceil) = -tk \log \frac{p}{k} + O(k). \tag{15}$$

Using the fact that $k = o(p)$ completes the proof of the upper bound.

We now prove the lower bound. By (12),

$$\widetilde{\mathrm{P}}_p^{\otimes 2}[\langle \mathbf{x}, \mathbf{x}'\rangle \geq t] \geq p(\lceil tk \rceil),$$

and combining this with (14) yields the claim. $\qquad\square$