[Reviews · NeurIPS 2020]

Review 1

Summary and Contributions: This paper studies the sharp phase transition phenomenon in the Gaussian additive model. A near-orthogonality condition is proposed under which the all-or-nothing phenomenon always holds. Then authors use sparse tensor PCA as an application to illustrate the effectiveness of the findings.

Strengths: The paper is theoretically sound. The theory improves the existing works in terms of the sparsity level.

Weaknesses: The content of this work seems to be on the light side. It would improve the quality of the paper if authors can study more machine learning applications to show the importance of the main theorem. In addition, it would be good to include some numerical simulations to validate the theoretical findings.

Correctness: I believe them to be correct.

Clarity: The paper is well written and easy to read.

Relation to Prior Work: The paper gives a nice review of and comparison with existing works.

Reproducibility: Yes

Additional Feedback:


Review 2

Summary and Contributions: The paper consider the problem of estimating a rank-1 sparse tensor of any order d>=2 in the presence of Gaussian additive noise. It shows that when the signal is Bernoulli or Rademacher then the sparse tensor PCA problem exhibits the "all-or-nothing phenomenon". This curious phenomenon means that it is statistically impossible to get constant non-zero correlation with the signal below a certain SNR, and above this SNR it is possible to get perfect correlation with the signal. To show this result for the tensor PCA problem, the authors study an interesting problem of estimating a signal with a discrete prior in the presence of Gaussian additive noise. They show that the all-or-nothing phenomenon follows from a near-orthogonality property of vectors sampled from the discrete prior.

Strengths: 1. The paper considers a basic problem, and shows interesting results about it. I quite like the Gaussian additive model, it is something many people should be able to understand and appreciate. 2. The paper also appears to be able to develop a more general machinery for showing these results, which could perhaps be useful beyond this work. 3. The paper ends with an interesting set of open problems, particularly those related to computational efficiency.

Weaknesses: 1. One could make the argument that some of the key messages in the paper have been demonstrated to an extent in prior work. For instance, the problem has been studied in the regime of sparsity Omega(p) and sparsity Omega(p^{2/3}). The current work extends these results to any sparsity. 2. The authors express that their results are shown via a much simpler argument. I do think this is the case, and as mentioned above the Gaussian additive model setup is quite nice. However, I think the exposition of the technical part and the proof could be better. This would help the authors make a better case that their argument is cleaner and simpler. I discuss this in some more detail later.

Correctness: The paper appears to be technically sound.

Clarity: The first few pages of the paper mostly read well. However, about half the paper is spent on the proofs, and this part could be made much more accessible. I also think the authors should spend more space on Section 2. It would be helpful to provide more examples and intuition here, most readers will appreciate that more than the detailed proofs. For instance, maybe provide a some more examples of distributions which will satisfy the overlap property, and distributions which will not satisfy this property. Would M_N=exp(N) randomly sampled vectors from the unit sphere satisfy this? It appears not, since the rate function should be t^2 in that case. Should one expect the all-or-nothing phenomenon in this case? I think examples like this will help the reader develop intuition and interest. I also think the detailed proofs in Section 3 and 4 should be accompanied by informal high-level sketches.

Relation to Prior Work: Overall, the authors do a decent job at discussing prior work, but I have two remarks. One point is that in line 70 the authors say the results prove a conjecture implied in several prior works, it would be good to be a bit more specific here, to put the contribution in context. Also, I felt that it would be good to compare the techniques in the paper with the previous papers in some more detail, since one of the claims of the paper is that it offers a simpler argument.

Reproducibility: Yes

Additional Feedback: 1. If the techniques were extendable to maybe one more setup than the sparse tensor PCA problem then that would greatly strengthen the argument that the techniques are more general. 2. Could the all-or nothing phenomena be demonstrated for continuous priors? Is there a limitation on all the current techniques which prevents them for extending to the continuous case? 3. The authors mention that Banks et al. and Perry et al. show that constant correlation with the signal is possible above a certain SNR ("partial recovery"). Is "almost perfect" recovery also possible in the same regime, or is the result in these previous works tight in these regards? 4. Is Theorem 2 tight, if not, is there a simple counterexample? Also, r(t)>=3*t should probably also be a sufficient condition for the Theorem, it is perhaps simpler to understand it as it is linear. Some minor remarks: 1. Line 107, say that this is the Bayes-opt estimator. 2. Line 110, maybe remind the reader that X is normalized to unit norm and hence 1 is the worst possible MSE. -----Post author response------ I thank the authors for the feedback. The feedback clarified a mistake in my review, I thought that the all-or-nothing phenomenon were already known for sparse *tensor* PCA in certain sparsity regimes. I also appreciate the commitment of the authors to revise the writing, which would make the paper much stronger. Therefore, I my improving my score for the paper.


Review 3

Summary and Contributions: This paper studies the question of recovering a planted "structure" in the presence of additive Gaussian noise. This is an important class of signal recovery problems capturing examples such as sparse PCA and tensor PCA. The main question in such questions is determining the minimum signal strength required to (exactly or approximately) recover the added signal. This paper is focused on a certain "all-or-nothing" phenomenon - this means that there's a SNR threshold above which, the Bayes' optimal estimator achieves almost-perfect recovery while below it, almost nothing of the signal can be recovered. The framework seems general and appealing and is applied to obtain a threshold for the reasonably well-studied problem of sparse Tensor PCA. Prior works showed a similar phenomenon but proved a all-or-nothing type result for recovering a vector correlated with the additive "spike". This work shows the stronger result with almost-perfect recovery replacing the approximate one above.

Strengths: Appealing characterization of when all-or-nothing phenomenon holds for additive Gaussisan models. Improved result for sparse Tensor PCA.

Weaknesses: Proofs themselves are largely uninsighful and follow the standard outline present in prior works. IMO, the main contribution is the conceptual abstraction of a condition (in terms of KL distance to the "null" distribution) that governs the all-or-nothing phenomenon.

Correctness: Yes.

Clarity: Yes.

Relation to Prior Work: yes.

Reproducibility: Yes

Additional Feedback:

[Author Response · NeurIPS 2020]

We thank the reviewers for their positive feedback and insightful criticism. We first address some general comments appearing in more than one review and then proceed with addressing the additional comments of each reviewer.

**Clarity of the technical parts of the paper.** We agree with the reviewers that the proofs of our results in the submitted paper would greatly benefit by adding beforehand summaries of the high level ideas and proof sketches, at least for our main theorems. We commit on doing this for our revised version of our work. Specifically, upon acceptance, we plan on devoting part of the additional page for the camera-ready version of the paper to adding high-level proof summaries. We do agree with multiple reviewers that the arguments used are, for the most part, rather simple and indeed such an adjustment will hopefully make this (potentially appealing) aspect of this work more clear.

**Applications of our results beyond sparse tensor PCA** We thank the reviewers for asking whether our results for the Gaussian additive model can be applied to other models, beyond sparse tensor PCA. We first wanted to highlight that the reason we decided to focus solely on our application on sparse tensor PCA was simplicity; this is an extremely well-studied inference problem which is also easy to state, and it gives a clear example of the applicability of our main theorem to obtain tight results for all $k = o(p)$ and all $d \geq 2$.

As part of our ongoing work, we have also proven that our results in the submitted paper do imply the all-or-nothing phenomenon also for the $k$-Gaussian submatrix localization problem (a Gaussian version of the well-studied $k$-stochastic block model) [Banks et al. '18] when $\omega(1) = k = o(n^{\frac{1}{4}})$. We strongly believe that our method can be applied to establish the all-or-nothing phenomenon for various well-studied Gaussian additive models in the literature.

**Techniques: simplicity and novelty** Our second-moment-method techniques are indeed similar to ones existing in the literature (e.g. [Banks et al. '18, Perry et al. '20]); however, these works do not address the all-or-nothing nature of the recovery threshold. By contrast, our techniques are quite different from the statistical-physics-inspired tools of [Barbier et al '20]. We consider our ideas "simpler" as we do not attempt to fully characterize the limiting free energy of the model, but instead prove the all-or-nothing phenomenon directly via our characterization combined with the second-moment method. We commit to making this comparison clearer in our revision. We agree emphatically with Reviewer 5: our main aim in this paper is to present a clean and sharp condition equivalent with the all-or-nothing property. Once the statement of the theorem is guessed, the proofs follow the standard outline.

**Reviewer 1.**

**Numerical simulations** We thank the reviewer for suggesting numerical simulations to validate the theoretical findings. Unfortunately, such simulations are in our context prohibitive for a potentially fundamental reason, namely, that sparse tensor PCA is widely conjectured to be computationally hard at the information-theoretic threshold; that is, polynomial-time methods seem to require a much larger $\lambda_N$ to work. As a corollary, simulating the performance of the Bayes-optimal estimator is potentially only possible with exponential-time methods around the critical $\lambda = 2 \log M_N$.

**"The content of this work seems to be on the light side"** As mentioned by Reviewer 5, the main contribution of this paper is to establish that the all-or-nothing phenomenon is equivalent to an explicit condition on the Kullback-Leibler divergence. In that sense, the goal of this paper is conceptual simplicity.

**Reviewer 3.**

**Key messages demonstrated in prior work for sparsity** $k = \Omega(p^{2/3}), k = o(p)$ We respectfully disagree with the reviewer that the all-or-nothing phenomenon for sparse tensor PCA for sparsity $k = \Omega(p^{2/3}), k = o(p)$ has been demonstrated in previous work. This has been done solely in the sparse *matrix* PCA case, i.e. when $d = 2$, in [Barbier, Macris '20]. In our result we establish it for all $k = o(p)$ and any $d \geq 2$.

**More examples in Section 2** We wholeheartedly agree with the reviewer that adding more examples below our theorems would help the reader develop important intuition about our results. The reviewer also asked the interesting question of whether $M_N$ uniform random points from the sphere would satisfy the overlap condition of Theorem 2. While we are not sure if this is the case, we can prove that as long as $M_N \to +\infty$, the all-or-nothing phenomenon nevertheless still holds holds for this example with high probability, by directly bounding the KL divergence and using Theorem 1. We thank the reviewer for raising this nice example, and we plan to add it to a revision to emphasize the applicability of Theorem 1.

**Tightness of Theorem 2** We thank the reviewer for this question. Theorem 2 is not tight, as one can show directly via Theorem 1 that the all-or-nothing phenomenon holds for sparse tensor PCA when $d = 1$, yet the overlap condition fails. We plan to add this comment after the theorem in the revised version of our paper, to help the intuition of the reader.

[Meta-Review · NeurIPS 2020]

This paper gives a pleasing general condition for the "all or nothing" phenemonon around the SNR for signal recovery in additive Gaussian models. Specifically, this paper shows that sparse tensor PCA exhibits he "all or nothing" phenomenon. That is, there's a threshold SNR below which, nothing of the sparse added spike can be recovered while above which, almost everything can be recovered. Beyond the actual results, the paper gives a simple and intuitively interpretable sufficient condition based on the KL divergence between two distributions that governs the all or nothing phenomenon. This paper studies an important class of statistical signal recovery models and gives an elegant insight into when and what kind of recovery is possible. I am pleased to recommend acceptance to NeurIPS.